# Effect of methotrexate use on the development of type 2 diabetes in rheumatoid arthritis patients: A systematic review and meta-analysis

**Leena R. Baghdadi** *

Department of Family and Community Medicine, College of Medicine, King Saud University, Riyadh, Saudi Arabia

* lbaghdadi@ksu.edu.sa

## Abstract

### Background

The high risk of cardiovascular disease is well recognized in rheumatoid arthritis. Type 2 diabetes also attributes to this increase in risk. Rheumatoid arthritis is a chronic inflammatory condition, which aggravates insulin resistance, placing the patients at a higher risk of type 2 diabetes and subsequent cardiovascular outcomes. Methotrexate treatment, as a gold standard anti-inflammatory drug in the treatment of rheumatoid arthritis has shown beneficial effects on cardiovascular health. However, its impact on type 2 diabetes is still unknown.

### Objective

To assess the strength of the association between exposure to methotrexate and the rate of development of type 2 diabetes in rheumatoid arthritis patients.

### Methods

All rheumatoid arthritis studies reporting the use of methotrexate as an exposure and type 2 diabetes as an outcome were searched until March 2020 using MEDLINE, Cochrane and Scopus databases. Studies were included if the diagnosis of rheumatoid arthritis was made according to current guidelines or by a rheumatologist, and if there was information about methotrexate exposure and the type 2 diabetes outcome. The author and an independent assessor evaluated the articles for eligibility. Meta-analyses combined relative risk estimates from each study where raw counts were available.

### Results

Sixteen studies reporting sufficient data for inclusion in the meta-analyses were identified. Methotrexate showed a promising effect on the risk of type 2 diabetes as this risk decreased in rheumatoid arthritis patients using methotrexate (Relative risk 0.48, 95% CI 0.16, 1.43).

**Data Availability Statement:** All relevant data are within the paper and its Supporting Information files.

**Funding:** The author received no specific funding for this work.

**Competing interests:** The author has declared that no competing interests exist.

## Conclusion

Rheumatoid arthritis patients on methotrexate treatment had a lower risk of developing type 2 diabetes compared to rheumatoid arthritis patients not exposed to methotrexate. This finding highlights the need for future, randomized control trials to confirm the beneficial effect of methotrexate on type 2 diabetes in the rheumatoid arthritis population.

## Introduction

Rheumatoid arthritis (RA) is a chronic inflammatory disease. It is usually complicated by local joint deformities and extra-articular manifestations including the renal, respiratory and cardiovascular (CV) systems [1]. Undeniably, CV disease is the most common and serious complication associated with RA [2]. Excess risk of CV morbidity and mortality among RA patients was recorded at 48% [3] and 60% [4], respectively. The burden of CV disease is increasing worldwide [5]. The excess CV risk among RA patients was argued to be triggered by the coexistence of systemic inflammation [6–9].

Until now, there is no definite evidence of the role of inflammation associated with RA and other CV risk factors in CV disease mortality and morbidity. However, a growing body of evidence demonstrates that the risk of developing CV disease increased with the presence of any of the traditional CV risk factors; which include high blood pressure, type 2 diabetes (T2D), obesity, hypercholesterolemia, smoking and physical inactivity [10]. This association indicates that the CV risk might, at least partially, be explained by the presence of CV risk factors. A former systematic review [10] reported that CV risk increased among RA patients with hypertension, T2D, smoking, hypercholesterolemia and obesity. The impact of T2D in increasing the CV risk among the RA population has been reported by several studies [11–13]. Diabetic RA patients had almost twice the risk of developing CV disease compared to nondiabetic RA patients (Relative risk [RR] 1.94, 95% CI 1.58, 2.30) [10]. This higher CV risk among RA patients with T2D has been reported by several observational studies [14–16]. Moreover, a higher risk of CV events was also reported in diabetic RA patients compared to the general population [15, 17].

RA is associated with chronic inflammation, which might, at least in part, trigger the development of T2D [18]. Thus, T2D might be a sequel of RA, which increases the risk of CV disease in this population. Although there is no clear mechanism of how T2D increases the CV risk in inflammatory diseases, there is emerging evidence that a substantial increase in insulin resistance (IR) associated with rheumatic diseases might play a part [19, 20]. A recent randomized controlled trial showed that conventional disease-modifying antirheumatic drugs (cDMARDs), mainly methotrexate (MTX) and infliximab, decrease IR. This improvement was greater among patients using MTX and infliximab together (42% reduction, p = 0.003) compared with other treatment groups [21]. Another study reported a similar reduction in the IR of patients using MTX for 3 months; however, the relationship did not reach statistical significance, which could be explained by the short duration of exposure to MTX [22].

Recent evidence suggests that MTX, an analogue of B-vitamin, folic acid, is the gold standard cDMARD used for treating RA patients, and exerts additional beneficial effects on CV health [23]. Emerging evidence suggests that the use of MTX might lower not only the overall risk of CV disease [24] but also the risk of individual CV risk factors such as hypertension [25]. RA patients commenced on MTX were found to have significantly lower peripheral and central blood pressure compared to those not using MTX. Although there are inconsistencies in

the effect of MTX on other CV risk factors [19, 26], some studies have shown that the prevalence of T2D was reduced among patients using DMARDs [20, 21]. However, there is lack of conclusive evidence about this protective effect. Therefore, the aim of this systematic review and meta-analysis was to investigate the relative impact of MTX exposure on the development of T2D among RA patients by reviewing the literature systematically and carrying out a meta-analysis of eligible studies. Prevention of T2D development might decrease the risk of developing CV disease and in turn reduce associated mortalities.

## Material and methods

### Methodology

A literature search was conducted for all articles including abstracts published in English, and about the relationship between MTX and T2D among patients with RA. Pre-Medline, Medline, Cochrane and Scopus databases were searched until March 2020; the Pre-Medline and Medline databases were searched using PubMed. Controlled vocabulary terms (MeSH terms) and keywords were used to identify articles. The following search terms were used: "rheumatoid" or the MeSH terms "rheumatoid arthritis", in combination with "methotrexate" or "anti-rheumatic", combined with "diabetes" or "type 2 diabetes" (details in S3 Materials). The citation lists were handsearched for relevant articles in an effort to look for additional papers. There is no existing protocol for this systematic review.

### Study selection and patient outcomes

The exposure of interest was RA patients on MTX monotherapy. The comparison group was RA patients not taking MTX (either MTX naïve, taking other antirheumatic drugs, i.e. cDMARDs, or biological disease-modifying antirheumatic drugs [bDMARDs]). MTX exposure was defined as taking an MTX drug for at least 8 weeks. The primary outcome was the rate of T2D, defined by the presence or absence of T2D. Per patient data was calculated as the number of RA patients that had T2D out of the total group studied over the follow-up period.

Articles were independently evaluated for eligibility by the author and an independent assessor (assistant professor and senior researcher, King Saud University). Inclusion criteria for studies in the meta-analyses were the diagnosis of RA in adult patients (≥18 years) made by a rheumatologist or according to the current RA guidelines (the European League Against Rheumatism [EULAR]/ or the American College of Rheumatology [ACR]); documentation of MTX exposure; assessment of the outcome of interest (T2D), and reported raw count data. Relevant studies were excluded, if this inclusion criteria were not fulfilled; no information about MTX exposure was available; no information about the outcome (T2D) was available, and the required raw count data was not available.

### Data extraction

Data from each study was summarized in terms of the study design, participant characteristics, assessed MTX exposure, details of non-MTX antirheumatic medications, assessed outcome, assessment of other traditional CV risk factors and study quality score (Qi).

### Quality scores of included studies

Included studies have different study designs with a variety of methodological approaches. Thus, considering the quality of the pooled studies was essential. A validated and reproducible checklist (S4 Materials) was used to critically evaluate selected studies; this tool was feasible and effective in differentiating papers with high precision and less bias from poor quality

studies [27, 28]. It also enabled calculating a quality score (Qi) for each study. This checklist examines the quality of each study using 14 questions to evaluate the internal validity, external validity and statistical analysis of the study (S4 Materials) [27]. Each question in the checklist was given points to calculate the Qi score. One of the questions in the checklist accounting for prognostic factors (question 9 in S4 Materials) was tailored to accommodate the requirements of this study. The prognostic score was created to balance the indicators affecting T2D outcomes across exposure groups. These prognostic factors include the age, sex, hypertension, body mass index (BMI), dyslipidemia, family history of T2D and CV disease, physical inactivity, duration of RA, and medications used (folic acid, corticosteroids and cDMARDs and bDMARDs). If the study balanced ≥5 of these indicators among comparison groups, it was given a score of 1. On the other hand, if the study reported 3 or 4 indicators, it was given a score of 0.5; if the study balanced 1 or 2 of these factors or if there was no evidence of reporting any of these prognostic factors, it was given a score of 0. After critically reading each study and giving points for each question, the total points were summed up to obtain the Qi score. A Qi score of ≥10 was defined as a high-quality score and a Qi score ≤9 was defined as a low-quality score.

## Statistical analysis

The association between exposure to MTX and development of T2D was assessed by conducting meta-analyses using MetaXL software version 5.3 (EpiGear International, Brisbane, Queensland, Australia). A quality-effects model (QE) was used for obtaining the meta-analyses and a random-effects model (RE) was used for comparison purposes. The QE model has an advantage over the RE model as it considers the quality of included studies by adjusting for the study-level risk of bias [29].

For the T2D outcome, raw counts for exposed and nonexposed, and event and nonevent RA groups were used to calculate the effect sizes and confidence intervals (CI). Incidental or prevalent T2D was considered an event. As the outcome of interest is binary (with T2D or without T2D), RR was used to measure the association between the exposure to MTX and the development of T2D in the RA population. While an RR of 1 indicates equivalence, an RR value <1 indicates that there is a reduced risk of developing T2D in RA patients exposed to MTX compared to RA patients not taking MTX.

Testing heterogeneity with statistical methods such as the Q-statistic and its variants has low statistical power. Therefore, adding another approach to the statistical methods is required when synthesizing the results of different studies. This additional approach includes applying prior biological knowledge, being vigilant and using common sense when testing for heterogeneity [30]. The presence of statistical heterogeneity was expected among study groups if the Q-statistic was significant (p <0.1) and/or tau-squared was less than zero.

The $I^2$ statistic was used to assess statistical heterogeneity. This test quantifies the percentage of total variation across included studies, which resulted from heterogeneity rather than chance; a substantial level of heterogeneity is reported if the value of $I^2$ is 50% or more [31]. When heterogeneity was documented, potential influential factors were assessed including the length of follow-up, age, methods of MTX assessment, methods of T2D assessment, usage of antirheumatic medications, number of recruited RA patients and the percentage of included males compared to females. In addition, the effect of MTX exposure on the development of T2D among RA patients was further analyzed by performing a subgroup analysis. The probable modifying effects of different characteristics of RA patients included in the study were examined. These include age, duration of RA, length of follow-up, status of T2D, RA disease activity measured by the Disease Activity Score 28 (DAS-28), year and country of publication.

Subgroups were defined as mean age of RA patients, ≤60 years or >60 years; mean RA duration, ≤2 years or >2 years; time of follow-up, ≤5 years or >5 years; status of T2D, incident or prevalent T2D; RA disease activity measured by DAS-28, measured disease activity or unmeasured disease activity; year of publication, before or in 2010 or after 2010; country of publication, North America (the United states of America [USA]), Latin America, Asia Pacific and Europe. Potential publication bias was assessed by examining asymmetry in the funnel plot, which is created by plotting the effect measure against the inverse of its standard error, and with Egger's test of the intercept [32]. The presence of asymmetry and p <0.05 indicate statistically significant publication bias.

## Results

### Search and screening

There were 4,310 studies found through searching electronic databases and an additional published study identified through handsearching. After removal of duplicates, there were 4,170 studies published between December 1974 and March 2020. After screening the abstracts of these studies, 4,076 were excluded as they did not meet the inclusion criteria, leaving a total of 94 studies (Fig 1). The full-texts of the remaining 94 articles were evaluated for eligibility. Seventy-eight studies were excluded because 75 articles had no information about MTX exposure and T2D, one publication was written in German and was difficult to translate, and two studies had no data for RA.

Therefore, 16 studies that met the inclusion criteria, were included in the qualitative and quantitative analyses [11, 12, 14–17, 19, 25, 33–40]. There were five studies, which directly assessed the incident T2D in the RA population (16, 34, 35, 37, 40); however, these studies reported the effect of DMARDs (other than MTX) on T2D. Most included studies considered T2D as one of the CV risk factors in RA (n = 10). In one of the included studies, the occurrence of T2D was part of metabolic syndrome in RA patients commenced on DMARDs [36]. These studies were thoroughly examined to obtain appropriate data about MTX exposure and T2D outcomes in RA patients. After careful assessment of included studies, raw data for exposed and nonexposed groups, and event and nonevent groups were extracted from the 16 studies. They provided data on a total of 89,676 RA patients exposed to MTX and a total of 70,385 RA patients not exposed to MTX.

### Characteristics of studies and subjects

Specific information from the eligible 16 studies included in this systematic review and meta-analysis is described in Table 1. These studies vary considerably in study design, methodological quality, patients' age, disease duration, use of antirheumatic medications and ascertainment of the MTX exposure and outcome. All eligible articles were observational studies. Three studies had a cross-sectional design, one had a nested case-control design, eight were prospective cohorts and four were retrospective cohorts. The country of publication differed between the included studies. There were six studies conducted among RA patients in North America (the USA) [16, 19, 33, 34, 37, 40], two studies in Latin America (Mexico, Colombia, and Brazil) [12, 14], two in Asia Pacific (Australia and Taiwan) [25, 35] and six in Europe (France, Sweden and Netherlands) [11, 15, 17, 36, 38, 39].

There was a noticeable variation in the methodological quality of included studies; the quality score (Qi score) ranged from 7 [17] to 12 [25] (S5 Materials). Additionally, the mean age of RA patients ranged between 49 [35] and 67 years [38]. Patients included in this systematic review had established RA (mean range of RA duration was 2.2±8.7 [36] and 14.4±12.4 [37] years). Although early RA (mean RA duration 3.3 months) was examined in one of studies

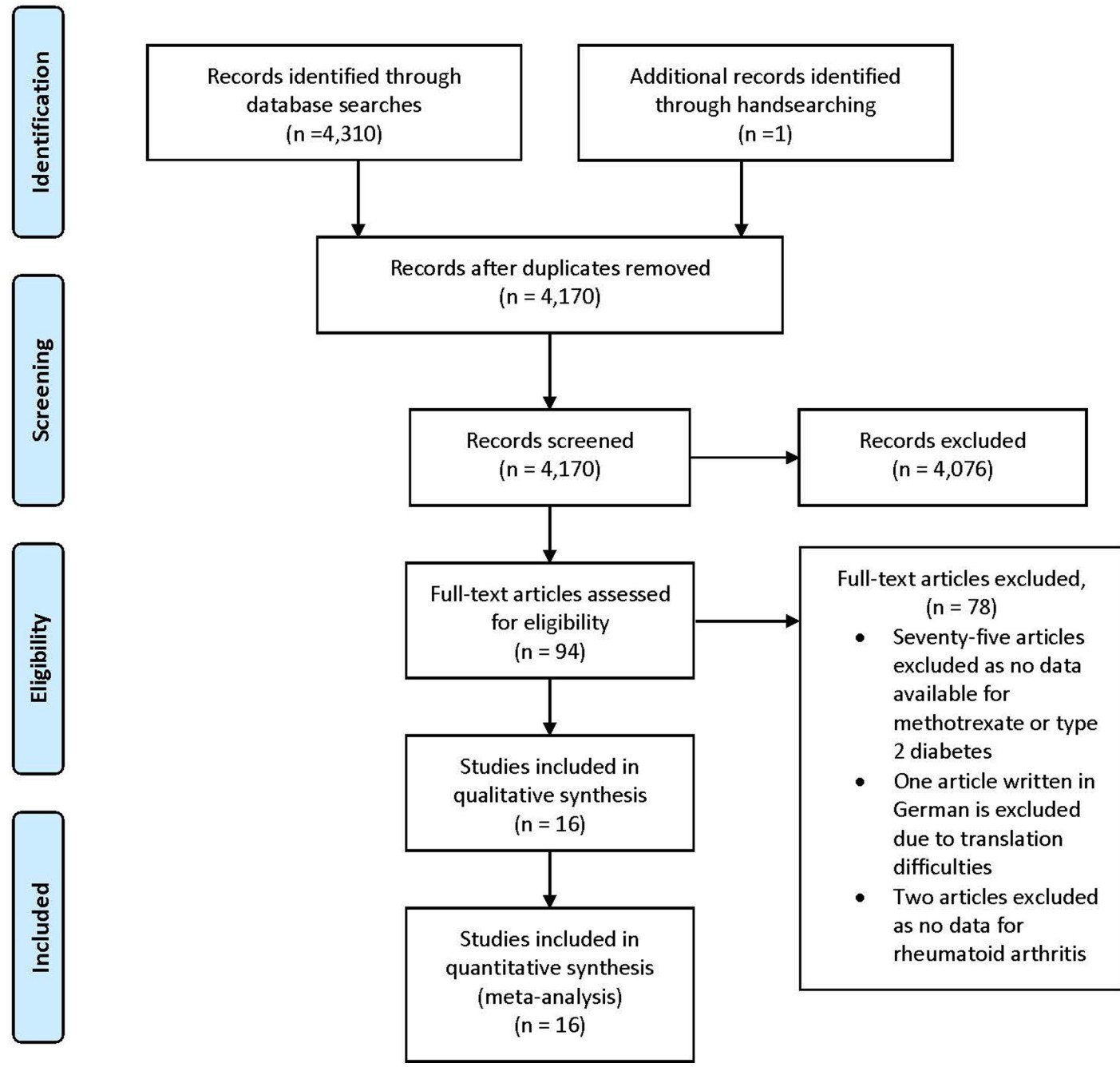

**Fig 1. Flow diagram of the studies eligible for the review and meta-analysis.**

[11], data were obtained during the follow-up period (5 years), which indicates that patients had established RA. Each study reported exposure to MTX, and the development of T2D in RA patients. However, no standard criteria were used when selecting the RA patients [37], ascertaining exposure to MTX [15, 39], or the occurrence of T2D [17, 25, 37, 38, 40]. Five studies recruited RA patients with incident T2D [16, 19, 34, 37, 40] while other studies included RA patients with prevalent T2D [11, 12, 14, 15, 17, 25, 33, 35, 36, 38, 39]. All included studies reported the use of DMARDs and provided details of the drug names except one study [12].

**Table 1. Overview of all included studies (n = 16), sorted by publication date in descending order.**

| Reference | Number of patients | Design | Country | Participants | MTX exposure assessed | Non- MTX DMARDs assessed | T2D outcome | Other traditional CV risk factors assessed | Qi score |
|---|---|---|---|---|---|---|---|---|---|
| **Agca et al., 2019** | Total RA patients 326 (Females 212; Males 114) | PCS | Amsterdam, the Netherlands | RA diagnosed according to the 1987 ACR classification criteria; mean age: 63±7 years; mean RA duration: 7±3 years; median follow-up time: 15 years; disease activity: measured by DAS-28 | Exposure to MTX was assessed as one of the DMARDs in RA patients and diabetics; there was no information about the status of using MTX, its dose or duration | cDMARDs: sulfasalazine, hydroxy-chloroquine and leflunomide. bDMARDs: no information about the drug names | T2D was one of the assessed traditional CV risk factors; it was defined by the 1999 WHO criteria of glucose ≥7.0 mmol/L (≥126 mg/dL) or treated with glucose-lowering medications | Hypertension, hypercholesterol-emia, smoking, BMI, family history of CVD, age and sex | 10.5 |
| **Ruscitti et al., 2019** | Total RA patients 841 (Females 691; Males = 150) | PCS | Italy | RA diagnosed according to ACR/EULAR criteria for RA; median age: 60 years; median RA duration: 8.2 years; median follow-up time: 3 years; disease activity: measured by DAS-28 | Exposure to MTX was assessed; no information about the status, dose or duration of using MTX | cDMARDs: hydroxy-chloroquine, sulfasalazine and leflunomide. bDMARDs: TNFα inhibitors | T2D was one of the assessed traditional CV risk factors; T2D defined by the ADA criteria or the use of antidiabetic medications | Hypertension, hypercholesterol-emia, smoking, BMI, age and sex | 11 |
| **Best et al., 2018** | Total RA patients 44,190 (Females 33,809; Males 10,381) | PCS | USA | RA diagnosed according to the ICD-9-CM; mean age: 58.9±13.4 years; median follow-up time: 1 year; disease activity: DAS-28 was not measured | Exposure to MTX was assessed as one of the DMARDs in RA patients; the dosage of all the medications was based on the NDC; no information about the status of using MTX, its dose or duration | cDMARDs: hydroxy-chloroquine, leflunomide and sulfasalazine. bDMARDs: etanercept, infliximab and adalimumab | Prevalence of T2D was assessed; it was defined based on the ICD-9-CM code 250. X | Age and sex | 10 |
| **Gomes et al., 2018** | Total RA patients 338 (Females 307; Males = 31) | Cross-sectional study | Fortaleza, Brazil | RA diagnosed according to the ACR/EULAR 2010 criteria for RA; mean age: 53.5±12 years; mean RA duration: 2.2±8.7 years; median follow-up time: 1 year; disease activity: measured by DAS-28 | Exposure to MTX was assessed as current use of the medication; there was no information about the dose or duration of MTX use | cDMARDs: leflunomide. bDMARDs: no information about the drug names | T2D was assessed as one of the components of metabolic syndrome. T2D defined by fasting blood glucose >100mg/dL or if the patient was taking any antidiabetes medications | Hypertension, hypercholesterol-emia, smoking, BMI, family history of CVD, physical inactivity, age and sex | 7.5 |

(*Continued*)

**Table 1.** (Continued)

| Reference | Number of patients | Design | Country | Participants | MTX exposure assessed | Non- MTX DMARDs assessed | T2D outcome | Other traditional CV risk factors assessed | Qi score |
|---|---|---|---|---|---|---|---|---|---|
| **Chen et al., 2017** | Total RA patients 33,112 | RCS | Taiwan | RA diagnosis based on the ICD9 code 714.0 after at least 3 outpatient clinic visits and on holding a catastrophic illness certificate; mean age: 49.2 ±14.9 years; median follow-up time: 12 years; disease activity: DAS-28 was not measured | Exposure to oral or injectable MTX was assessed; no information on the status of using MTX, its dose or duration | cDMARDs: leflunomide, sulfasalazine, azathioprine, cyclosporine. bDMARDs: anti-TNF agents | T2D diagnosed based on the ICD-9-CM code 250.X and the concurrent prescription of any antidiabetes medications | Age and sex | 9.5 |
| **Mangoni et al., 2017** | Total RA patients 86 (Females 62; Males 24) | Repeated cross-sectional study | Australia | RA diagnosed according to the 1987 ACR or the 2010 ACR/ EULAR criteria for RA; mean age: 61±13 years; mean RA duration: 9±10.8 years; follow-up period: 8 months; disease activity: measured by DAS-28 | Exposure to MTX was assessed as current use of MTX; defined as using MTX for ≥8 weeks; median MTX dose: 15 mg per week; median MTX duration: 75 months | cDMARDs: hydroxy-chloroquine, leflunomide and sulfasalazine. bDMARDs: abatacept, rituximab, tocilizumab, adalimumab, etanercept, certolizumab pegol and golimumab | T2D was one of the assessed traditional CV risk factors; its definition based on physician's diagnosis or if patient reported treatment with glucose-lowering medications at the time of assessment | Hypertension, hypercholesterol-emia, smoking, BMI, family history of CVD, physical inactivity, age and sex | 12 |
| **Ozen et al., 2017** | Total RA patients 13,669 (Females = 10,953; Males = 2,716) | PCS | USA | RA was diagnosed based on rheumatologist's diagnosis; mean age: 58.6±13.4 years; mean RA duration: 14.4 ±12.4 years; median follow-up time: 4.6 years; disease activity: DAS-28 was not measured | Exposure to MTX was assessed as ever use of MTX; there was no information about the MTX dose or duration | cDMARDs: hydroxy-chloroquine. bDMARDs: TNFα inhibitors, abatacept, rituximab and tocilizumab | Incident T2D based on patients' self-report of diagnosing new T2D or initiating antidiabetic medications | Hypertension, smoking, BMI, age and sex | 11 |
| **Amaya-Amaya et al., 2013** | Total RA patients 800 (Females 647; Males 153) | Cross-sectional | Colombia | RA diagnosed according to the 1987 ACR classification criteria for RA; mean age: 51.8 ±12.1 years; mean RA duration: 12.4 ±10.3 years; study period: between February 1996 and April 2012; disease activity: measured by DAS-28 | Exposure to MTX was assessed as current or past use of MTX in the RA population; there was no information about the status of using MTX, its dose or duration | cDMARDs: sulfasalazine, D-penicillamine, gold salts, leflunomide, azathioprine and cyclosporine. bDMARDs: etanercept, infliximab, adalimumab, abatacept, tocilizumab and rituximab | T2D was one of the assessed traditional CV risk factors; definition based on fasting glucose ≥7.0 mmol/L (126 mg/dL) or if treated with glucose-lowering medications at the time of assessment | Hypertension, hypercholesterol-emia, smoking, BMI, family history of CVD, abdominal obesity, physical inactivity, age and sex | 9 |

(*Continued*)

**Table 1.** (Continued)

| Reference | Number of patients | Design | Country | Participants | MTX exposure assessed | Non- MTX DMARDs assessed | T2D outcome | Other traditional CV risk factors assessed | Qi score |
|---|---|---|---|---|---|---|---|---|---|
| Antohe et al., 2012 | Total RA patients 1,587 (Females 1,150; Males 437) | PCS | Central Pennsylvania, USA | RA diagnosed according to the 1987 ACR classification criteria for RA; median age: 57 years; median follow-up time: 3 years; disease activity: DAS-28 was not measured | Exposure to MTX was assessed as continuous use of the medication; continuous use was defined as a discontinuation gap of <3 months; the start and the stop dates of using MTX were recorded during the observation period; there was no information about the dose of MTX | cDMARDs: hydroxy-chloroquine bDMARDs: TNFα inhibitors including adalimumab, etanercept, golimumab and infliximab | Incident T2D was the primary outcome; it was defined based on the physician-established diagnosis or using the 2010 ADA criteria, random glucose level of ≥200 mg/dl, HbA1c ≥6.5%, or ever use of any hypoglycemic or antidiabetic medications. | BMI, age and sex | 9 |
| Bili et al., 2011 | Total RA patients 1,127 (Females 823; Males 304) | RCS | Central Pennsylvania, USA | RA diagnosed according to the 1987 ACR classification criteria for RA; mean age: 60.7 ±15.1 years; median follow-up time: 2 years; disease activity: DAS-28 was not measured | Exposure to MTX was assessed as continuous use of the medication; continuous use was defined as a discontinuation gap of <3 months; the start and stop dates of using MTX were recorded during the observation period; there was no information about the dose of MTX | cDMARDs: hydroxy-chloroquine. bDMARDs: TNFα inhibitors | Incident T2D based on the 2010 ADA criteria: random glucose ≥200 mg/dL or HbA1c ≥6.5% or ever use of hypoglycemic/ antidiabetic medications | BMI, age and sex | 10 |
| Innala et al., 2011 | Total patients at entry 700 (Females 481; Males 219;); Total patients at end 442 (Females 301; Males 141;) | PCS | Sweden | Early RA diagnosed by ARA criteria; patient records and self-reported questionnaire on comorbidity and local rheumatologist follow-up assessments were used; mean age: 55.2±14.3 years; mean disease duration: 3.3 months; median follow-up time: 5 years; disease activity: measured by DAS-28 | Exposure to ever use of MTX was assessed; accumulated number of months of treatment was considered, there was no information about the MTX duration or its dose | cDMARDs: sulfasalazine, chloroquine, azathioprine, mycopheno, atmophetil, myocrisine, auranofin, cyclosporine and leflunomid. bDMARDs: etanercept, adalimumab, infliximab, anakinra and rituximab | T2D was one of the assessed traditional CV risk factors; there was no standardized definition for T2D, it was assessed based on patients' self-reported questionnaire on comorbidity | Hypertension, hypercholesterol-emia, smoking, BMI, age and sex | 11 |

(*Continued*)

**Table 1.** (Continued)

| Reference | Number of patients | Design | Country | Participants | MTX exposure assessed | Non- MTX DMARDs assessed | T2D outcome | Other traditional CV risk factors assessed | Qi score |
|---|---|---|---|---|---|---|---|---|---|
| **Solomon et al., 2011** | Total RA patients 11,327 (Females 8,900; Males = 2,427) | PCS | USA | RA diagnosed based on at least 2 visits to the rheumatologist using the ICD-9-CM 714.x criteria; mean age: 53.7±13.2 years; median follow-up time: 5.8 months; disease activity: DAS-28 was not measured | Exposure to oral or injectable MTX was assessed; there was no information about the MTX status, dose or duration of use | cDMARDs: hydroxy-chloroquine, sulfasalazine, leflunomide, cyclosporine, azathioprine, cyclo-phosphamide, mycophenolate mofetil, 6-thioguanine, acitretin, D-penicillamine, gold, auranofin, myochrysine, and solganol; bDMARDs: TNFα inhibitors including dalimumab, etanercept an infliximab | Incident T2D defined by presence of at least 1 diagnosis of T2D based on ICD-9-CM250.x, and a new user of T2D-specific medications including all insulin preparations and oral antidiabetic agents | Age and sex | 10 |
| **Radovits et al., 2009** | Total RA patients 222 (Females 145; Males 77) | Nested case-control study | Nijmegen, the Netherlands | RA diagnosed according to the 1987 revised ACR criteria; random selection of controls from the PCS; mean age: 67.5±10 years; mean RA duration: 4.3±8.5 years;; disease activity: measured by DAS-28 | Exposure to MTX was assessed among RA cases and controls; cumulative MTX dose among RA cases and controls: 1791.6 vs. 1943.9 mg; total MTX treatment duration among RA cases and controls: 152.9 vs. 161.9 weeks | bDMARDs: TNFα inhibitors | T2D was one of the assessed traditional CV risk factors; there was no standardized definition for T2D | Hypertension, hypercholesterol-emia, smoking, BMI, family history of CVD, age and sex | 11 |
| **Assous et al., 2007** | Total RA patients 239 (Females 196; Males 43) | RCS | France | RA diagnosed according to the 1987 revised ACR criteria; mean age: 56.3±15.7 years; mean RA duration: 11.6±8.8 years; mean follow-up time: 5.4±1.8 years; disease activity: DAS-28 was not measured | Exposure to MTX was assessed as one of the DMARDs in RA patients; duration of use ≥1 month; there was no information about the status of using MTX or its dose | bDMARDs: infliximab, etanercept, and adalimumab | T2D was one of the assessed traditional CV risk factors; there was no standardized definition for T2D | Hypertension, hypercholesterol-emia, smoking, BMI, age and sex | 7 |

(*Continued*)

**Table 1.** (Continued)

| Reference | Number of patients | Design | Country | Participants | MTX exposure assessed | Non- MTX DMARDs assessed | T2D outcome | Other traditional CV risk factors assessed | Qi score |
|---|---|---|---|---|---|---|---|---|---|
| **Wasko et al., 2007** | Total RA patients 4,905 | PCS | USA | RA diagnosed according to the 1987 ACR revised classification criteria for RA; mean age: 58.2 ±13.9 years; mean RA duration: 13 ±11.7 years; median follow-up time: 21.5 years; disease activity: DAS-28 was not measured | Exposure to MTX was assessed as ever or never use of MTX; there was no information about the MTX dose, duration of MTX use (percentage of observation time): 39% | cDMARDs: hydroxy-chloroquine | Incident T2D based on patients' self-report of the diagnosis or the new use of hypoglycemic medications | Age and sex | 10 |
| **del Rincon et al., 2001** | Total RA patients 236 (Females 146; Males 90) | RCS | Mexico | RA diagnosed according to 1987 ACR criteria; O´RALE cohort used; matched non-RA: SAHS cohort; median age: 56 years (range 22–80); median follow-up: time 7 years; disease activity: DAS-28 was not measured | Exposure to MTX was current use of the medication. It was one of the DMARDs assessed in RA patients; there was no information about the duration of using MTX or its dose | No information about DMARDs (MTX exposure, dose, or duration) | T2D was one of the assessed traditional CV risk factors; it was defined based on the 1997 criteria of the ADA: glucose ≥7.0 mmol/L (≥126 mg/dL) or if treated with glucose-lowering medications at the time of assessment | Systolic blood pressure, hypercholesterol-emia, cigarette smoking, BMI, age and sex | 11 |

ACR = American College of Rheumatology; ADA = American Diabetes Association; ARA = American Rheumatism Association; bDMARDs = biological disease-modifying antirheumatic drugs; BMI = body mass index; CV = cardiovascular; CVD = cardiovascular diseases; DAS-28 = Disease Activity Score 28; EULAR = American College of Rheumatology/European League Against Rheumatism; HbA1c = glycosylated hemoglobin; ICD-9-CM = International Classification of Diseases, 9th Revision, Clinical Modification; MTX = methotrexate; NDC = national drug codes; O´RALE = Outcome of Rheumatoid Arthritis Longitudinal Evaluation (del Rincon et al 2001); PCS = prospective cohort study; Qi = quality score; RA = rheumatoid arthritis; RCS = retrospective cohort study; SAHS = San Antonio Heart Study cohort; T2D = type 2 diabetes; cDMARDs = conventional disease-modifying antirheumatic drugs; TNFα = tumor necrosis factor alpha; USA = United States of America; WHO = World Health Organization.

Traditional CV risk factors other than T2D were assessed in 12 studies [11, 12, 14–17, 25, 34, 36–39], namely, hypertension, hypercholesterolemia, smoking, BMI or/and obesity, family history of CV disease, physical inactivity, age and sex. However, four studies [19, 33, 35, 40] assessed only the age and sex of the RA patients (i.e. non-modifiable CV risk factors).

## Risk of type 2 diabetes

All 16 studies were included in the meta-analyses. The effect size was in favor of a promising beneficial effect of MTX in reducing the risk of T2D in RA patients. The QE model RR for all these studies was 0.48 (95% CI 0.16, 1.43), implying that the risk of T2D among RA patients exposed to MTX was reduced by 52% compared to those RA patients not taking MTX (Fig 2). A similar effect was found by the RE model RR 0.13 (95% CI 0.08, 0.22), suggesting that there

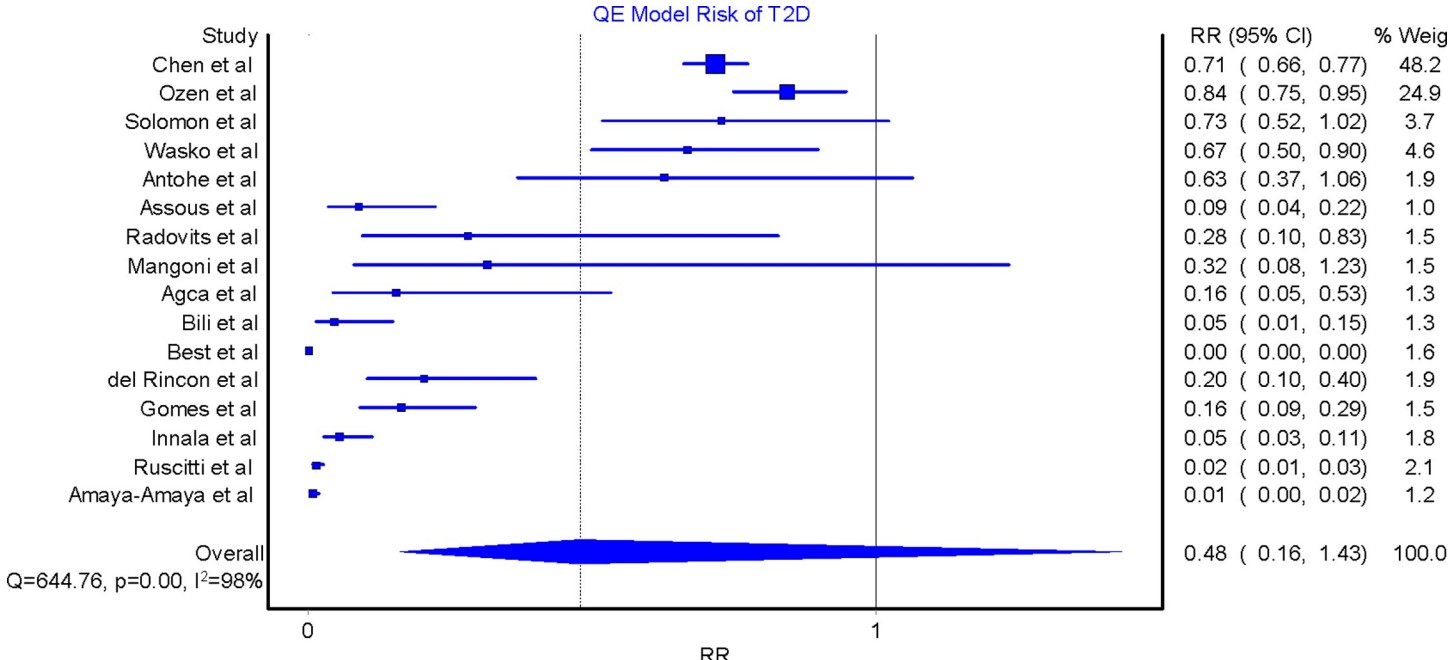

**Fig 2. Relative risk of type 2 diabetes in rheumatoid arthritis patients on methotrexate versus those not taking methotrexate.** QE = quality effects model; RR = relative risk; T2D = type 2 diabetes.

was an 87% reduction in the risk of T2D among RA patients taking MTX compared to those who were not taking MTX (Fig 3).

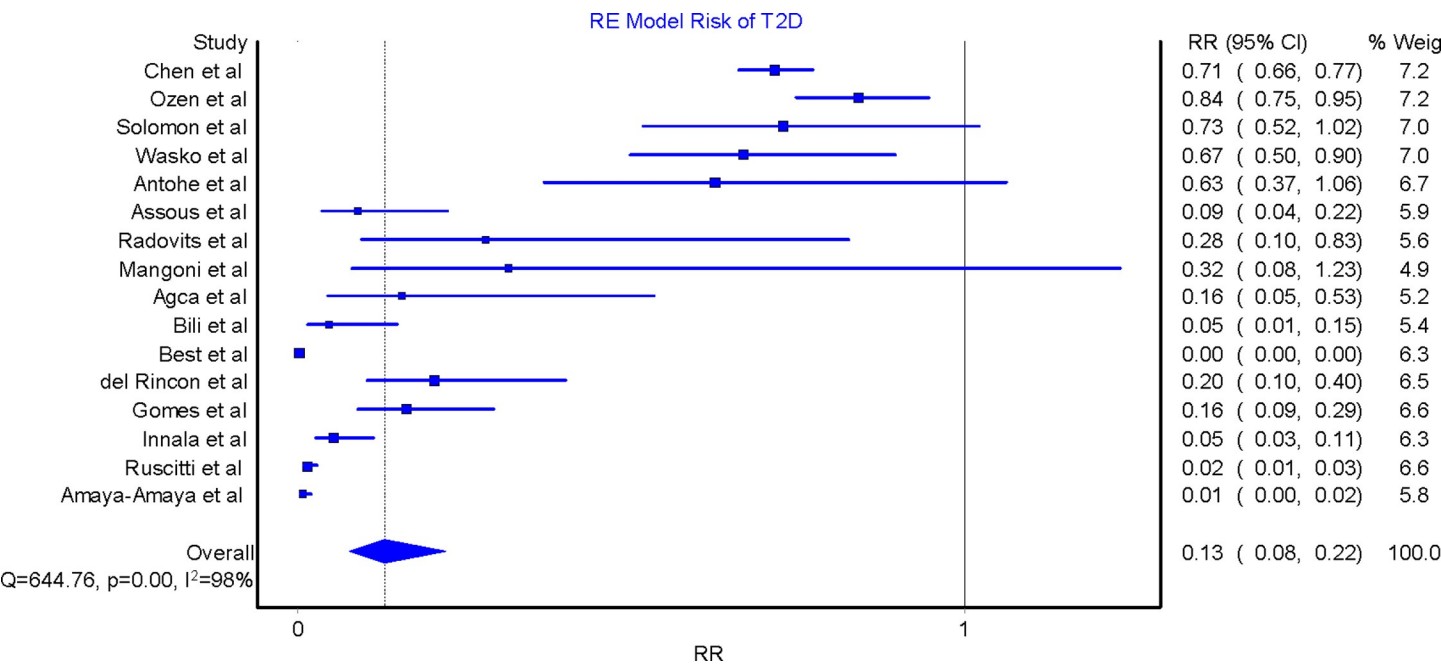

**Fig 3. Relative risk of type 2 diabetes in rheumatoid arthritis patients on methotrexate versus those not taking methotrexate.** RE = random effects model; RR = relative risk; T2D = type 2 diabetes.

Sensitivity analysis (Table 2) examined the possible effects of various factors deemed to influence T2D outcomes and assessed the degree of sensitivity of the obtained results to such changes. Possible factors, which might influence the T2D outcomes, were considered in the analysis. The pooled RR was recalculated after subgrouping the mean age of RA patients ($\leq$60 years or >60 years), mean RA duration ($\leq$2 years or >2 years), time of follow-up ($\leq$5 years or >5 years), status of T2D (incident or prevalent T2D), RA disease activity measured by DAS-28 (measured disease activity or unmeasured disease activity), year and country of publication (before or in 2010 and after 2010). Results presented in Table 2 show that RA patients taking MTX had greater reduction in the development of T2D if they were older (>60 years); the duration of RA was shorter ($\leq$2 years); the follow-up time was longer (>5 years); they had prevalent T2D and were reported in studies published before 2010; the RA disease activity was measured and adjusted in the study.

Comparison of results after subgrouping the studies into different regions showed that the risk of T2D among MTX users decreased more in patients recruited for studies conducted in Latin America and Europe compared to those from North America (USA) and the Asia Pacific. In addition, a sensitivity analysis was carried out after excluding the study by Best et al. (2018) as its effect estimate showed an RR of zero. The pooled results for the T2D outcome were not significantly different from the original results; both RE and QE models showed a reduction in the T2D outcome among RA patients on MTX, reporting an RR of 0.19 (95% CI 0.12, 0.29) and an RR of 0.52 (95% CI 0.32, 1.21), respectively. Nevertheless, there was no significant variation in these results compared to the original findings. There is evidence of

**Table 2. Sensitivity analyses of methotrexate and type 2 diabetes in RA patients.**

| Parameters and type 2 diabetes [RR QE model (95% CI)] |
| --- |
| **Mean age of subjects (years)** |
| $\leq$60 (n = 9): **0.61** (95% CI 0.23, 1.61) |
| >60 (n = 7): **0.05** (95% CI 0.01, 0.26) |
| **Mean duration of RA (years)** |
| $\leq$**2 (n = 2): 0.09** (95% CI 0.03, 0.27) |
| >**2 (n = 8): 0.57** (95% CI 0.06, 4.96) |
| **Time of follow-up (years)** |
| $\leq$**5 (n = 7): 0.57** (95% CI 0.05, 6.69) |
| >**5 (n = 6): 0.51** (95% CI 0.03, 9.42) |
| **Status of T2D** |
| **Incident T2D (n = 5): 0.77** (95% CI 0.43, 1.37) |
| **Prevalent T2D (n = 11): 0.37** (95% CI 0.01, 9.95) |
| **Disease activity** |
| **Measured disease actvity (n = 7): 0.06** (95% CI 0.02, 0.18) |
| **Unmeasured disease activity (n = 9): 0.61** (95% CI 0.24, 1.58) |
| **Year of publication** |
| $\leq$**2010 (n = 4): 0.45** (95% CI 0.12, 1.64) |
| >**2010 (n = 12): 0.47** (95% CI 0.14, 1.59) |
| **Country of publication** |
| **North America (USA) (n = 6): 0.67** (95% CI 0.09, 5.00) |
| **South America (n = 2): 0.08** (95% CI 0.00, 2.81) |
| **Asia Pacific (n = 2): 0.65** (95% CI 0.37, 1.13) |
| **Europe (n = 6): 0.07** (95% CI 0.02, 0.18) |

n = number of studies involved in the analysis, USA = United States of America

publication bias suggested by the funnel plot of the included 16 studies (S1 Fig); this finding is confirmed by Egger's test, which shows significant asymmetry (p <0.05).

## Discussion

To the best of our knowledge, this systematic review and meta-analysis is the first to explore the impact of MTX on the development of T2D in the RA population. It provides evidence for a significant positive effect of this drug on decreasing the risk of developing T2D.

High CV risk in the RA population is well documented. T2D is a well-known risk factor for CV disease not only in the general population [41], but also in the RA population. RA patients with T2D had almost twice the risk of developing CV disease compared to nondiabetic RA patients (RR 1.94, 95% CI 1.58, 2.30) [10]. This higher CV risk among diabetic RA patients has been reported by many observational studies [14–16]. This two-fold increase in the CV risk among diabetic RA patients is similar to the risk observed in the general population [15, 17]. This indicates that T2D, as one of the traditional CV risk factors, increases the CV risk independent of inflammation. About 50% reduction in cardiovascular risk (RR 0.41, 95% CI 0.15, 0.96) was observed with a modest improvement in the glycemic status among T2D patients [42]. Hence, one of the recommendations introduced by the EULAR guidelines for CV risk management in patients with RA, is aggressive treatment to lower the inflammation associated with RA, and to modify the presence of CV risk factors such as T2D [43].

Although there is no conclusive evidence about the effect of MTX on the development of T2D, results from studies included in this meta-analysis showed that MTX decreases the risk of T2D in RA patients (RR 0.48, 95% CI 0.16, 1.43). This finding is consistent with several studies that reported a reduction in the incidence rate of T2D among patients with rheumatic diseases and using other cDMARDs including hydroxychloroquine (HCQ) [44] and TNF inhibitors (TNFi) [45]. An observational study examined the relationship between MTX and the glycemic level [26]. The latter study reported a slight reduction in the level of glycosylated hemoglobin (HbA1c), among diabetic RA patients. However, the results from this study might be underestimated, as the MTX sample size was small and not precisely powered to detect differences in the HbA1c levels among MTX users. Similarly, a smaller reduction in the T2D incidence among MTX users compared to other cDMARDs was reported by a larger observational study [19]. Once again, this finding might be underestimated and might be exposed to misclassification bias as about two-thirds of the RA cohort had been previously exposed to MTX drugs; however, it was classified with a non-MTX cohort. Thus, the greater reduction in T2D rates observed among other cDMARD users might be, at least partially, explained by the prior MTX exposure.

In line with the finding of this meta-analysis, MTX has shown a positive impact not only on T2D but also on other CV risks in the RA population [24, 25, 46, 47]. In contest to our findings, however, an observational study claims that the risk of T2D was higher among RA patients using MTX compared to other DMARDs (RR 1.46, 95% CI 1.26, 1.68) [37]. Similarly, the cardioprotective effect of low-dose MTX was not documented among patients with inflammatory diseases such as atherosclerosis [48]. The author of this study, has reported in previous observational data that the use of low-dose MTX had repeatedly shown an association with reduced vascular event rates in patients with RA [48]. This paradox might have resulted from different types of selection bias that occurred predominantly in research about rheumatic diseases. This form of bias has been termed 'index event bias' or 'collider stratification bias' [49–51], and occurs as a result of conditioning on a common effect (the index event). This leads to an induction of spurious (perhaps opposite) association between the exposure and outcome. Therefore, this MTX paradox might be caused by index event bias in which MTX is associated

with the improvement of the disease (RA 'index event') and the RA disease sequelae (T2D); resulting in conditioning of the index event of RA [50]. This conditioning may lead to a spurious association (incorrect effect estimate) between the exposure and outcome due to unknown or uncontrolled confounders associated with both, the index event and the events downstream of RA. One of the possible confounders in the study [37] is the use of other anti-inflammatory drugs. T2D risk reduction was observed among MTX users when exposure to other DMARDs was adjusted in the analyses, indicating that MTX monotherapy has a positive impact in decreasing the development of T2D among RA patients.

Potential factors that might influence the pooled results were studied in the subgroup analysis. A relatively lower risk of T2D was observed in patients aged >60 years, RR 0.05 compared to those ≤60 years of age, with RR 0.61. This lower risk among older RA patients could be explained by the longer duration of exposure to MTX. This finding is consistent with the results of a prospective cohort study, showing older RA patients using MTX had better survival outcomes and reduced cardiovascular mortality [52]. Similarly, a trend towards a decreased T2D risk was documented in RA patients with longer follow-up periods (>5 years). It is possible that this observed reduction in T2D was due to the longer duration of MTX exposure. As most of the RA patients have established RA and have been exposed to MTX for a long duration, it is possible that the observed reduction in the T2D risk is related, at least partially, to the long MTX exposure. This was supported by the decrease in T2D among prevalent cases of T2D compared to the incident cases of T2D. Although the dose of MTX is clinically important, we found in our previous studies that the concentration of MTX inside the red blood cells (RBCs) (i.e. MTX polyglutamate [MTXPG] concentration) is more accurate [25, 46]; as approximately 95% of the MTX dose is metabolised within 24 hr of administration. Additionally, these studies [25, 46] show that the RBC MTXPG concentration is influenced not only by genetic but also non-genetic factors. In line with this finding, increased age and a longer duration of MTX use were associated with higher MTX concentrations [53]. This study found another major non-genetic determinant of increased MTXPG concentration, which is a lower estimated glomerular filtration rate (GFR). Lower GFR is common among the elderly with chronic diseases, such as T2D and RA, and use of prednisolone and nonsteroidal anti-inflammatory drugs (NSAIDs). These drugs are commonly used by RA patients and induce the inhibition of prostaglandin production [54]. A longer duration of exposure to MTX [50] might also explain the lower risk, which decreased by about 50% among prevalent T2D patients compared with incident T2D patients (RR 0.77 vs. 0.37). Moreover, our finding of lower risk of T2D in patients with a shorter RA duration is consistent with other studies [35]. RA patients taking MTX, with a disease duration ≤2 years (i.e. early RA) had a greater reduction in their T2D risk (RR 0.09, 95% CI 0.03, 0.27). This reduction in the risk might be attributed to the beneficial effect of MTX as patients with early RA are usually commenced on MTX monotherapy as a first line treatment. Those RA patients with a good initial response to MTX continue to have excellent 2-year clinical outcomes [55] and improved CV risks [25]. Additionally, it is possible that many diabetic RA patients who had longer disease durations had undiagnosed T2D that went unmanaged. This explanation is supported by published evidence indicating that RA has been associated with a higher prevalence of undiagnosed T2D, particularly in patients with longer RA durations [56]. Additionally, it is well known that patients with long-standing RA had greater disease activity and severity, which is related to a higher risk of T2D [36]. This study found that RA patients taking MTX had greater significant reduction in the development of T2D, if the disease activity was measured and adjusted in the analysis (RR 0.06, 95% CI 0.02, 0.18). This indicates that MTX has an independent beneficial effect on T2D development in the RA population. The risk of T2D among MTX users was reduced further in patients recruited for studies conducted in Latin America and Europe compared to those from

North America (USA) and Asia Pacific. The higher risk of T2D in the USA and Asia Pacific region might be partially explained by the differences in the social determinants of health influenced by geography, epidemiology of chronic diseases such as obesity, and type of lifestyle [57].

The pooled RR of one of the studies [33] was zero, implying that the exposure factor (MTX) might be an extremely protective factor [58]. There were seven cases of T2D out of the total MTX group (n = 20,873), indicating that T2D as an outcome is extremely rare among MTX users [33]. Regardless of that evidence, an additional sensitivity analysis was carried out without the study [33]. In this meta-analysis, the pooled result for T2D was still in favor of MTX exposure; both RE and QE models showed a reduction in the T2D outcome among RA patients on MTX drugs reporting RR 0.19 (95% CI 0.12, 0.29) and RR 0.52 (95% CI 0.32, 1.21), respectively.

Although there are variations in the reported evidence about the impact of DMARDs on T2D, there is emerging evidence supporting the favorable effect of HCQ [44], MTX [26] and TNFi [45] on altering glucose metabolism. The underlying mechanisms of T2D risk reduction with HCQ and TNFi are not fully established. However, one of the explanations, which is independent of the anti-inflammatory effect of HCQ and TNFi, is related to the improvement in insulin sensitivity and pancreatic β-cell functions, which decrease the risk of T2D [59, 60]. To our knowledge, the mechanism of MTX in improving IR and glucose homeostasis is still debatable. Nevertheless, emerging evidence shows that MTX reduces the level of HbA1c in RA and psoriasis [61]; after 6 months of MTX therapy, although the level of IR was unchanged, the concentration of HbA1c was reduced from 5.80 ± 0.29% to 5.51 ± 0.32%, p < 0.001. Even though the exact mechanism of such an improvement is not fully understood, MTX seems to offer long-lasting beneficial metabolic effects beyond its anti-inflammatory actions [37, 46, 61]. TNFi was found to have a long-lasting clinical effect on T2D, which lasted up to 6 months after cessation of the drug [37]. We, therefore, hypothesize that MTX improves the level of HbA1c and glucose homeostasis as the MTXPG inhibit 5-aminoimidazole-4-carboxamide ribonucleotide formyltransferase. The consequent accumulation of the substrate aminoimidazole carboxamide ribonucleotide (AICAR) and its metabolites inhibit adenosine deaminase and adenosine monophosphate (AMP) deaminase [62,63]. This leads to a rise in the intracellular concentrations of AICAR-monophosphate (ZMP) and AMP [64]. ZMP activates the adenosine monophosphate (AMP)-activated protein kinase (AMPK). Activation of AMPK ameliorates glucose homeostasis in T2D [65, 66]. Therefore, MTX may ameliorate glucose homeostasis by promoting ZMP accumulation and AMPK activation [67, 68]. Similar to metformin, which is an antihyperglycemic agent [69], MTX is able to decrease hepatic production and intestinal absorption of glucose and increase insulin sensitivity via the activation of AMPK.

This systematic review and meta-analysis has some limitations. Controlling for corticosteroid exposure was one of the limitations; although few RA patients had been exposed at some point of management to a small dose of prednisolone (i.e. ≤5 mg per day), there was no detailed information about the exposure. The EULAR task force concluded that a low dose of the glucocorticoid, prednisone (≤5 mg per day) is generally safe for RA patients [43]. However, there is no conclusive evidence about the effect of low dose corticosteroids (≤5mg per day) on developing T2D in the RA population. Most published studies reported a transient increase in blood glucose levels after taking a high to moderate dose of prednisolone, which usually resolves after drug discontinuation [43, 70]. Additionally, the duration of RA plays a vital role in RA management with corticosteroids. Commencing RA patients on corticosteroids is especially recommended by the ACR in early RA; based on the 2015 ACR and 2019 EULAR guidelines for the treatment of RA. Patients with early RA are usually started on low-

dose corticosteroids (i.e. ≤5 mg of prednisolone or its equivalent); early RA is defined by the ACR as a disease duration ≤6 months [43, 71]. In this study, however, most of the RA patients were diagnosed with established RA as the mean RA duration ranged between 2.2±8.7 and 14.4±12.4 years.

Moreover, there is an indication for publication bias. Large studies and positive findings were published more frequently compared to small studies; usually, authors do not submit insignificant or negative results for publication. Additionally, papers with such findings are often rejected by reviewers and/or editors leading to publication bias. There was also a variation in the participants' age at enrolment, duration of RA, duration of follow-up, country and time of the study.

Regardless of these limitations, numerous strengths should also be acknowledged. The impact of these factors (possible confounders affecting the effect estimates) was further examined by subgroup analysis that retained the trend of a lower risk of T2D in RA patients exposed to MTX. Considering the Qi score of included studies when calculating the effect estimates is another strength of this meta-analysis.

## Conclusion

In conclusion, our meta-analysis indicates that MTX shows a promising effect in reducing the risk of T2D in the RA population. Short- and long-term effects of MTX on the development of not only T2D but also other CV risk factors need to be explored by controlled clinical trials. In addition, the effects of MTX can be compared to other DMARDs to confirm its CV protective effects and establish possible mechanisms. If this association is proven by conducting more randomized clinical trials, MTX could be used as a cardioprotective and/or antidiabetic drug in the RA population as they are at higher risk of CV morbidities and mortalities. Thus, careful recognition and diagnosis of RA patients at a higher risk of T2D is mandatory, to initiate appropriate management and reduce the risk of T2D and associated CV risk. Future experimental and human randomized clinical trials are needed to investigate the association between MTX and CV risks.

## Supporting information

**S1 Fig. Funnel plot meta-analysis type 2 diabetes using fixed effect model (Legend: FE = fixed effect model; RR = relative risk; T2D = type 2 diabetes).**
(TIF)

**S1 Materials. PRISMA 2009 flow diagram.**
(DOC)

**S2 Materials. PRISMA 2009 checklist.**
(DOC)

**S3 Materials. Appendix 1.** Data sources.
(DOCX)

**S4 Materials. Appendix 2.** Checklist for assessing quality of included studies in the meta-analysis.
(DOCX)

**S5 Materials. Appendix 3.** Quality assessment scores for studies included in the meta-analysis.
(PDF)

**S1 Dataset.**
(PDF)

## Acknowledgments

Special thanks for support by the College of Medicine Research Center, Deanship of Scientific Research, King Saud University Riyadh, Saudi Arabia. Many thanks to Dr Emad Mahmoud, assistant professor and senior researcher at King Saud University, for evaluating the eligibility of the articles for inclusion in this systematic review and meta-analysis.

## Author Contributions

**Conceptualization:** Leena R. Baghdadi.

**Data curation:** Leena R. Baghdadi.

**Formal analysis:** Leena R. Baghdadi.

**Investigation:** Leena R. Baghdadi.

**Methodology:** Leena R. Baghdadi.

**Project administration:** Leena R. Baghdadi.

**Resources:** Leena R. Baghdadi.

**Validation:** Leena R. Baghdadi.

**Writing – original draft:** Leena R. Baghdadi.

**Writing – review & editing:** Leena R. Baghdadi.

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
