## [Decision Letter · Decision Letter 0]

13 May 2020

PONE-D-20-08145

Effect of methotrexate use on the development of type 2 diabetes in rheumatoid arthritis patients: a systematic review and meta-analysis

PLOS ONE

Dear dr Baghdadi ,

Thank you for submitting your manuscript to PLOS ONE. After careful consideration, we feel that it has merit but does not fully meet PLOS ONE’s publication criteria as it currently stands. Therefore, we invite you to submit a revised version of the manuscript that addresses the points raised during the review process.

Particularly the methodological aspects, indicated by all three reviewers, need attention

We would appreciate receiving your revised manuscript by Jun 27 2020 11:59PM. To enhance the reproducibility of your results, we recommend that if applicable you deposit your laboratory protocols in protocols.io, where a protocol can be assigned its own identifier (DOI) such that it can be cited independently in the future. For instructions see: http://journals.plos.org/plosone/s/submission-guidelines#loc-laboratory-protocols

We look forward to receiving your revised manuscript.

Kind regards,

Michael Nurmohamed, MD, PhD

Academic Editor

PLOS ONE

Reviewers' comments:

Reviewer's Responses to Questions

**Comments to the Author**

1. Is the manuscript technically sound, and do the data support the conclusions?

Reviewer #1: Partly

Reviewer #2: Partly

Reviewer #3: Yes

2. Has the statistical analysis been performed appropriately and rigorously? 

Reviewer #1: Yes

Reviewer #2: Yes

Reviewer #3: I Don't Know

3. Have the authors made all data underlying the findings in their manuscript fully available?

Reviewer #1: Yes

Reviewer #2: Yes

Reviewer #3: Yes

4. Is the manuscript presented in an intelligible fashion and written in standard English?

Reviewer #1: Yes

Reviewer #2: Yes

Reviewer #3: Yes

5. Review Comments to the Author

Reviewer #1: This is a very original systematic review, it was a very hard work. It is an important issue whether MTX would decrease the risk of T2DM or not.

1. The main result is actually not significant as 95%CI is 0.16-1.43. This is also shown in the forest plots, therefore the main conclusion is that MTX may non-significantly decrease T2DM risk but it is not at all sure. Author needs to change wording.

2. Significant effects were found at age above 60 and at disease duration less than 8 years. Author should further elaborate why elderly patients wouzld have better responses to MTX in terms of T2DM. Also, why was 8 years chosen as a cutoff for disease duration, Early RA (duration <2 years or so) could have been compared with long-term RA. 8 years of diease duration is already way long-term.

3. Even if it is not on T2DM, author could have cited the striking study of Ridker et al (NEJM 380: 752, 2019) showing that low dose MTX does not reduce CV risk. Author could compare inflammatory state like RA to non-inflammatory state. Would MTX work differently under inflammatory (e.g. RA) and non-inflammatory conditions?

Reviewer #2: The paper shows some limitations.

First: only one author revised all of the documents? In general in order to perform a systematic review is necessary three experts.

Second: I suggest to conventional DMARDs instead of instead of tDMARDs

The inclusion and exclusion criteria need to be clarified, in particular the languages and if the abstracts are included and which? EULAR? ACR?

Reviewer #3: Dear authors,

The impact of methotrexate use on type 2 diabetes is a very relevant subject, as the authors already point out in their introduction. In this meta-analysis, exposure and endpoint are both well defined in the method section. However, the studies included in this meta-analysis are very heterogeneous, and most of these studies do not define the exposure to methotrexate or the outcome of T2D that well. To me it is not clear if the authors accounted for the heterogeneity in methotrexate use, for example high or low dose, or longer or shorter exposure. Furthermore, it is not clear how the analysis accounted for the use other DMARDS. The authors have performed a set of sensitivity analyses, but data on disease activity is not mentioned, although it is hypothesized that this has a great influence on the development of T2D.

6. PLOS authors have the option to publish the peer review history of their article (what does this mean?). If published, this will include your full peer review and any attached files.

Reviewer #1: No

Reviewer #2: No

Reviewer #3: Yes: M Heslinga

---

## [Author Response · Author response to Decision Letter 0]

26 May 2020

Response to the reviewers comments

All the references added to the manuscript in order to address the reviewers’ comments and improve the manuscript have been highlighted.

Reviewer #1: This is a very original systematic review; it was a very hard work. It is an important issue whether MTX would decrease the risk of T2DM or not.

1. The main result is actually not significant as 95%CI is 0.16-1.43. This is also shown in the forest plots, therefore the main conclusion is that MTX may non-significantly decrease T2DM risk but it is not at all sure. Author needs to change wording.

Thank you for highlighting this important point. Based on the Cochrane Handbook for Systematic Reviews, when there is a ‘positive’ but statistically non-significant trend, authors commonly describe this as ‘a promising’ effect.’(Schünemann HJ, Oxman AD, Vist GE, Higgins JPT, Deeks JJ, Glasziou P & Guyatt, GH. 2011. Chapter 12: Interpreting results and drawing conclusions. In: Higgins JPT & Green S (eds.) Cochrane Handbook for Systematic Reviews of Interventions. Version 5.1.0. The Cochrane Collaboration). Therefore, the wording for the results was updated.

Page 2, lines 41-42

Methotrexate showed a promising effect on the risk of type 2 diabetes as this risk decreased in rheumatoid arthritis patients using methotrexate (Relative risk 0.48, 95% CI 0.16, 1.43).

Page 17, lines 247-248

The effect size was in favor of a promising beneficial effect of MTX in reducing the risk of T2D in RA patients.

Page 26, lines 439-440

In conclusion, our meta-analysis indicates that MTX shows a promising effect in reducing the risk of T2D in the RA population

2. Significant effects were found at age above 60 and at disease duration less than 8 years. Author should further elaborate why elderly patients would have better responses to MTX in terms of T2DM. Also, why was 8 years chosen as a cut off for disease duration, Early RA (duration <2 years or so) could have been compared with long-term RA. 8 years of disease duration is already way long-term.

Thank you for your kind comments. Regarding the age, MTX exposure and reduced risk of T2D and relatively lower risk of T2D was observed in patients aged >60 years. This has been explained as:

Page 22, lines 341-361

This lower risk among older RA patients could be explained by the longer duration of exposure to MTX. This finding is consistent with the results of a prospective cohort study, showing older RA patients using MTX had better survival outcomes and reduced cardiovascular mortality [52]. Similarly, a trend towards a decreased T2D risk was documented in RA patients with longer follow-up periods (>5 years). It is possible that this observed reduction in T2D was due to the longer duration of MTX exposure. As most of the RA patients have established RA and have been exposed to MTX for a long duration, it is possible that the observed reduction in the T2D risk is related, at least partially, to the long MTX exposure. This was supported by the decrease in T2D among prevalent cases of T2D compared to the incident cases of T2D. Although the dose of MTX is clinically important, we found in our previous studies that the concentration of MTX inside the red blood cells (RBCs) (i.e. MTX polyglutamate [MTXPG] concentration) is more accurate [25, 46]; as approximately 95% of the MTX dose is metabolized within 24 hr of administration. Additionally these studies [25, 46] show that the RBC MTXPG concentration is influenced not only by genetic but also non-genetic factors. In line with this finding, increased age and a longer duration of MTX use were associated with higher MTX concentrations [53]. This study found another major non-genetic determinant of increased MTX concentration, which is a lower estimated glomerular filtration rate (GFR). Lower GFR is common among the elderly with chronic diseases, such as T2D and RA, and use of prednisolone and nonsteroidal anti-inflammatory drugs (NSAIDs). These drugs are commonly used by RA patients and induce the inhibition of prostaglandin production [54]. A longer duration of exposure to MTX [50] might also explain the lower risk, which decreased by about 50% among prevalent T2D patients compared with incident T2D patients (RR 0.77 vs. 0.37).

Pages 22-23, lines 363-367

RA patients taking MTX, with a disease duration ≤2 years (i.e. early RA) had a greater reduction in their T2D risk (RR 0.09, 95% CI 0.03, 0.27). This reduction in the risk might be attributed to the beneficial effect of MTX as patients with early RA are usually commenced on MTX monotherapy as a first line treatment. Those RA patients with a good initial response to MTX continue to have excellent 2-year clinical outcomes [55] and improved CV risks [25].

Thus, the cut off for disease duration was changed from 8 years to 2 years (early RA: ≤2 years and established RA: >2 years). The main result is that RA patients taking MTX had greater reduction in the development of T2D if the duration of RA was shorter (≤2 years); sensitive analysis methods (page 9, lines 177-178), results (page 18, lines 266, 270), table 2 (pages 18-19), and discussion (pages 22-23, lines 363-367) were updated accordingly. 

Page 9, lines 177-178

Subgroups were defined as mean age of RA patients, ≤60 years or >60 years; mean RA duration, ≤2 years or >2 years

Page 18, lines 266 and 270

…mean RA duration (≤2 years or >2 years),…

…the duration of RA was shorter (≤2 years);…

Pages 18-19 table 2

Mean duration of RA (years)

≤2 (n = 2): 0.09 (95% CI 0.03, 0.27) 

>2 (n = 8): 0.57 (95% CI 0.06, 4.96)

Pages 22-23, lines 363-367

RA patients taking MTX, with a disease duration ≤2 years (i.e. early RA) had a greater reduction in their T2D risk (RR 0.09, 95% CI 0.03, 0.27). This reduction in the risk might be attributed to the beneficial effect of MTX as patients with early RA are usually commenced on MTX monotherapy as a first line treatment. Those RA patients with a good initial response to MTX continue to have excellent 2-year clinical outcomes [55] and improved CV risks [25].

3. Even if it is not on T2DM, author could have cited the striking study of Ridker et al (NEJM 380: 752, 2019) showing that low dose MTX does not reduce CV risk. Author could compare inflammatory state like RA to non-inflammatory state. Would MTX work differently under inflammatory (e.g. RA) and non-inflammatory conditions?

Thank you for mentioning this paper. The main finding of the recommended Ridker study (reference 48) is now included in the manuscript (page 21, lines 322-326). 

Page 21, lines 322-326

Similarly, the cardioprotective effect of low-dose MTX was not documented among patients with inflammatory diseases such as atherosclerosis [48]. The author of this study, has reported in previous observational data that the use of low-dose MTX had repeatedly shown an association with reduced vascular event rates in patients with RA [24]. 

This randomised control trail (RCT) was not conducted among RA patients. RA is a chronic inflammatory condition associated with inflammatory disease flares. The management of RA patients includes aggressive treatment to control the inflammation and prevent further damage. MTX is the anchor drug to treat RA and almost all patients have been exposed to it at some stage of treatment. In fact, this RCT highlighted in the limitations that “our observational data had repeatedly shown an association of low-dose methotrexate use with reduced vascular event rates in patients with rheumatoid arthritis or psoriatic arthritis. The reported benefits in these observational studies may apply only to patients with existing systemic inflammatory conditions.” 

Therefore, comparing inflammatory and non-inflammatory conditions and examining if MTX works differently in both groups is out of the scope of this meta-analysis, but it is a great idea and can be examined separately by a systematic review and meta-analysis. 

 

Reviewer #2: The paper shows some limitations.

First: only one author revised all of the documents? In general, in order to perform a systematic review is necessary three experts.

Thank you for your valid comment. It is not a PRISMA requirement that a systematic review must have at least two authors, and the Cochrane Handbook For Systematic Reviews states that it does not "necessitate" it but encourages it. Despite these guidelines, for this systematic review, the articles were independently evaluated for eligibility by the author and an independent assessor (assistant professor and senior researcher). The assessor has been named and acknowledged in the manuscript.

Page 6, lines 113-114.

Articles were independently evaluated for eligibility by the author and an independent assessor (assistant professor and senior researcher at King Saud University).

Page 26, lines 454-456.

Many thanks to Dr Emad Mahmoud, assistant professor and senior researcher at King Saud University, for evaluating the eligibility of the articles for inclusion in this systematic review and meta-analysis. 

Nevertheless, this systematic review could still represent an appropriate methodological shortcut (Cochrane Handbook), if it is conducted by an experienced reviewer. The author is an expert clinical epidemiologist herself, who has published a couple of systematic reviews where she was the first author and did most of the hard work herself (95% of the final manuscript). One of these systematic reviews was published in 2015 and reached 144 citations.

Second: I suggest to conventional DMARDs instead of instead of tDMARDs

The inclusion and exclusion criteria need to be clarified, in particular the languages and if the abstracts are included and which? EULAR? ACR?

The manuscript says conventional DMARDs (cDMARDs). This change has been made and highlighted throughout the manuscript beginning with:

Page 5, line 77

conventional disease-modifying antirheumatic drugs (cDMARDs),

The inclusion and exclusion criteria have been updated. The articles and abstracts published in English were included in the study (page 5, lines 97-98) and the RA diagnosis was updated to include EULAR/ ACR criteria (page 6, lines 114-121).

Page 5, lines 97-98

A literature search was conducted for all articles including abstracts published in English, and about the relationship between MTX and T2D among patients with RA.

Page 6, lines 114-121

Inclusion criteria for studies in the meta-analyses were the diagnosis of RA in adult patients (≥18 years) made by a rheumatologist or according to the current RA guidelines (the European League Against Rheumatism [EULAR]/ or the American College of Rheumatology [ACR]); documentation of MTX exposure; assessment of the outcome of interest (T2D), and reported raw count data. Relevant studies were excluded, if this inclusion criteria were not fulfilled; no information about MTX exposure was available; no information about the outcome (T2D) was available, and the required raw count data was not available.

 

Reviewer #3: Dear authors,

The impact of methotrexate use on type 2 diabetes is a very relevant subject, as the authors already point out in their introduction. In this meta-analysis, exposure and endpoint are both well defined in the method section. 

1. However, the studies included in this meta-analysis are very heterogeneous, and most of these studies do not define the exposure to methotrexate or the outcome of T2D that well. To me it is not clear if the authors accounted for the heterogeneity in methotrexate use, for example high or low dose, or longer or shorter exposure. 

Regarding heterogeneity, the author looked for explanations of heterogeneity by conducting several sensitivity analyses (i.e. sub-group analyses) based on a priori hypothesis. Potential factors, which might be possible sources of heterogeneity, were carefully considered to diminish spurious false-positive findings; these factors were then analysed separately to see whether heterogeneity in those subgroups decreases.

Even though most of the included studies did not define the exposure to methotrexate (including the dose) or the outcome of T2D that well, these methodological flaws were considered in calculating the quality score of each study and fitted in the QR model (i.e. the pooled RR is calculated considering the quality score for all studies). This was explained in detail as follows:

Page 7, lines 128-146.

Included studies have different study designs with a variety of methodological approaches. Thus, considering the quality of the pooled studies was essential. A validated and reproducible checklist (S4 Materials) was used to critically evaluate selected studies; this tool was feasible and effective in differentiating papers with high precision and less bias from poor quality studies [27, 28]. It also enabled calculating a quality score (Qi) for each study. This checklist examines the quality of each study using 14 questions to evaluate the internal validity, external validity and statistical analysis of the study (S4 Materials) [27]. Each question in the checklist was given points to calculate the Qi score. One of the questions in the checklist accounting for prognostic factors (question 9 in S4 Materials) was tailored to accommodate the requirements of this study. The prognostic score was created to balance the indicators affecting T2D outcomes across exposure groups. These prognostic factors include the age, sex, hypertension, body mass index (BMI), dyslipidemia, family history of T2D and CV disease, physical inactivity, duration of RA, and medications used (folic acid, corticosteroids and cDMARDs and bDMARDs). If the study balanced ≥5 of these indicators among comparison groups, it was given a score of 1. On the other hand, if the study reported 3 or 4 indicators, it was given a score of 0.5; if the study balanced 1 or 2 of these factors or if there was no evidence of reporting any of these prognostic factors, it was given a score of 0. After critically reading each study and giving points for each question, the total points were summed up to obtain the Qi score. A Qi score of ≥10 was defined as a high-quality score and a Qi score ≤9 was defined as a low-quality score. 

Regarding the heterogeneity in methotrexate use, for example high or low dose, or longer or shorter exposure. The author specified the duration of exposure to MTX to be at least 8 weeks instead of the dose of MTX as stated on:

Page 6, lines 109-110.

MTX exposure was defined as taking an MTX drug for at least 8 weeks.

This is because the optimal clinical effect of MTX is usually seen after 8 weeks of commencing the drug. Most patients with rheumatoid arthritis are usually commenced on a low dose of methotrexate (i.e. ≤20mg), based on the EULAR guidelines (reference 71, Smolen et al. 2020). Although the dose of methotrexate is clinically important, we found in our previous studies that the concentration of methotrexate inside the RBCs (i.e. MTX polyglutamate [MTXPG] concentration) is more accurate (page 22, lines 349-354 and references 25 and 46); as approximately 95% of the MTX dose is metabolised within 24 hr of administration. 

Page 22, lines 349-354

Although the dose of MTX is clinically important, we found in our previous studies that the concentration of MTX inside the red blood cells (RBCs) (i.e. MTX polyglutamate [MTXPG] concentration) is more accurate [25, 46]; as approximately 95% of the MTX dose is metabolised within 24 hr of administration. Additionally, these studies [25, 46] show that the RBC MTXPG concentration is influenced not only by genetic but also non-genetic factors.

2. Furthermore, it is not clear how the analysis accounted for the use other DMARDS. The authors have performed a set of sensitivity analyses, but data on disease activity is not mentioned, although it is hypothesized that this has a great influence on the development of T2D.

Unfortunately, the status of disease activity was not measured in most of the included studies. While a couple of these studies measured Disease Activity Score DAS-28, most included studies have not considered it. In addition, the disease activity score could not be differentiated between treatment groups. Again, this methodological issue was considered when calculating the quality score and pooled RR (as stated in response to question 1, on page 7, lines 128-146).

Nevertheless, an additional sensitivity analysis was conducted to examine whether disease activity had an impact on the pooled results. The main finding was that RA patients taking MTX had greater reduction in the development of T2D, if the disease activity was measured and adjusted in the study. Information about disease activity was added in table 1 (pages 11-16), sensitive analysis methods (pages 8 and 9 lines 176-180), results (page 18 lines 267-268 and 272), table 2 (page 19), and discussion (page 23, lines 372-375). 

Table 1 can be found on pages 11-16 of the manuscript but has not been included in this response to reviewers as it is five pages long.

Pages 8 and 9, lines 176-177 and page 9, lines 179-180

…, RA disease activity measured by the Disease Activity Score 28 (DAS-28), year and country of publication. Subgroups were defined as mean age of RA patients, ≤60 years or >60 years; mean RA duration, ≤2 years or >2 years ; time of follow-up, ≤5 years or >5 years; status of T2D, incident or prevalent T2D; RA disease activity measured by DAS-28, measured disease activity or unmeasured disease activity;…

Page 18, lines 267-268 and 272

…RA disease activity measured by DAS-28 (measured disease activity or unmeasured disease activity)…

…the RA disease activity was measured and adjusted in the study.

Page 19, from table 2

Disease activity

Measured disease actvity (n= 7): 0.06 (95% CI 0.02, 0.18)

Unmeasured disease activity (n= 9): 0.61 (95% CI 0.24, 1.58)

Page23 line 372-375

This study found that RA patients taking MTX had greater significant reduction in the development of T2D, if the disease activity was measured and adjusted in the analysis (RR 0.06, 95% CI 0.02, 0.18). This indicates that MTX has an independent beneficial effect on T2D development in the RA population.

---

## [Decision Letter · Decision Letter 1]

19 Jun 2020

Effect of methotrexate use on the development of type 2 diabetes in rheumatoid arthritis patients: a systematic review and meta-analysis

PONE-D-20-08145R1

Dear Dr. Baghadi,

We’re pleased to inform you that your manuscript has been judged scientifically suitable for publication and will be formally accepted for publication once it meets all outstanding technical requirements.

Kind regards,

Michael Nurmohamed, MD, PhD

Academic Editor

PLOS ONE

Additional Editor Comments (optional):

Reviewers' comments:

Reviewer's Responses to Questions

**Comments to the Author**

1. If the authors have adequately addressed your comments raised in a previous round of review and you feel that this manuscript is now acceptable for publication, you may indicate that here to bypass the “Comments to the Author” section, enter your conflict of interest statement in the “Confidential to Editor” section, and submit your "Accept" recommendation.

Reviewer #1: All comments have been addressed

Reviewer #2: All comments have been addressed

2. Is the manuscript technically sound, and do the data support the conclusions?

Reviewer #1: Yes

Reviewer #2: Yes

3. Has the statistical analysis been performed appropriately and rigorously? 

Reviewer #1: Yes

Reviewer #2: Yes

4. Have the authors made all data underlying the findings in their manuscript fully available?

Reviewer #1: Yes

Reviewer #2: Yes

5. Is the manuscript presented in an intelligible fashion and written in standard English?

Reviewer #1: Yes

Reviewer #2: Yes

6. Review Comments to the Author

Reviewer #1: I have no further comments. The issues raised are addressed sufficiently. I have no ethical or other concerns.

Reviewer #2: The paper is interesting and well written. I suggest to accept the paper in the actual form.

I have not further comment.

7. PLOS authors have the option to publish the peer review history of their article (what does this mean?). If published, this will include your full peer review and any attached files.

Reviewer #1: No

Reviewer #2: No

---

## [Editor Report · Acceptance letter]

23 Jun 2020

PONE-D-20-08145R1 

Effect of methotrexate use on the development of type 2 diabetes in rheumatoid arthritis patients: a systematic review and meta-analysis 

Dear Dr. Baghdadi:

I'm pleased to inform you that your manuscript has been deemed suitable for publication in PLOS ONE. Congratulations! Your manuscript is now with our production department. 

Kind regards, 

on behalf of

Prof.Dr Michael Nurmohamed 

Academic Editor

PLOS ONE